# Risk factors for stillbirth and neonatal mortality among participants in Mobile WACh NEO pilot, a two-way SMS communication program in Kenya

Anna B. Hedstrom[1,2]*, Esther M. Choo[2], Keshet Ronen[2], Brenda Wandika[3], Wenwen Jiang[4], Lusi Osborn[3], Maneesh Batra[1,2], Dalton Wamalwa[5], Grace John-Stewart[1,2,4], John Kinuthia[3], Jennifer A. Unger[2,6]

1 Department of Pediatrics, University of Washington, Seattle, United States of America, 2 Department of Global Health, University of Washington, Seattle, United States of America, 3 Department of Research & Programs, Kenyatta National Hospital, Nairobi, Kenya, 4 Department of Epidemiology, University of Washington, Seattle, United States of America, 5 Department of Paediatrics & Child Health, University of Nairobi, Nairobi, Kenya, 6 Department of Obstetrics and Gynecology, University of Washington, Seattle, United States of America

* hedstrom@uw.edu

OPEN ACCESS

**Data Availability Statement:** The dataset is available in at https://github.com/keshetronen/neo_pilot_public.

## Abstract

Globally, 2.5 million neonates die and 2 million more are stillborn each year; the vast majority occur where access to life-saving care is limited. High quality, feasible interventions are needed to reach, educate and empower pregnant women and new mothers to improve care-seeking behaviors. Mobile WACh (Mobile solutions for Women's and Children's health) NEO is a human-computer hybrid mobile health (mHealth) system that allows for two-way short message service (SMS) communication between women and healthcare workers during the peripartum period. We performed a secondary prospective cohort analysis of data from the Mobile WACh NEO pilot study to determine maternal characteristics associated with neonatal death and stillbirth and examine participant messaging associated with these events. Pregnant women were enrolled at two Kenyan public health clinics between 28–36 weeks gestation. They received personalized, educational, action-oriented SMS messages during pregnancy and through 14 weeks postpartum. Participants could message the study at any time and study nurses responded. Standardized questionnaires assessed participant characteristics at baseline and 14 weeks postpartum. Outcomes were ascertained at study visits or by SMS report. Among 798 pregnant women enrolled, median age was 24 years [IQR 21, 29], 37% were primiparous and 92% used SMS as a primary mode of communication. Seventeen neonatal deaths and 13 stillbirths occurred. Older maternal age was associated with increased risk of stillbirth [aRR 1.12 (CI 1.02–1.24), $p$ <0.05]. We found no significant predictors of neonatal death. Participant messaging to study nurse about concerns in the week preceding death was less common prior to infant death after discharge home from facility birth (9%) than prior to stillbirth (23%). We found limited predictors of neonatal death and stillbirth, suggesting identifying women prenatally for targeted support may not be a feasible strategy. Scarce messaging from mothers whose neonates died may

**Funding:** Funding was provided by the following grants, listed with initials of authors who received funding from them: USAID (Saving Lives at Birth) AID-OAA-F-16-00026- KR, JK, JU, MB, DW, BW NIH/NICHD R01HD080460- KR, JK, WJ, JU, MB, GJS, BW NIH/NICHD 1R01HD098105- KR, JK, JU, DW, EC, AH, BW NIH/NICHD K24HD054314- GJS The funders had no role in study design, data collection and analysis, decision to publish, or preparation of the manuscript.

**Competing interests:** The authors have declared that no competing interests exist.

reflect difficulties identifying illness or rapid deterioration and needs to be better understood to design and test interventions for this high-risk period. Messaging prior to stillbirth, while at similar levels as other periods, does not appear to have an impact as most women do not experience identifiable signs or symptoms prior to the event.

## Introduction

Globally, 2.5 million neonates die each year and 2 million are stillborn, the vast majority in low and middle income countries (LMIC) where access to necessary care is limited for pregnant women and their newborns [1–3]. An estimated 3 million neonatal deaths and stillbirths are preventable with quality antenatal care, skilled delivery attendance and early neonatal care [4–6]. To address this gap in care, the Every Newborn Action Plan has targeted reaching every woman and newborn to reduce inequities in coverage and access to care [2, 6, 7]. Progress towards reduction in mortality has lagged in sub-Saharan Africa where delayed care seeking is a key factor limiting timely care [2, 8]. These delays increase risk of stillbirths and preventable neonatal deaths due to asphyxia, prematurity and infection [9]. In order to improve pregnancy and neonatal outcomes, high quality, feasible interventions are needed to reach, educate and empower pregnant women and new mothers to improve care-seeking behaviors.

The WHO recognizes that technological innovations are useful to improve care for mothers and newborns [2]. The wide use of mobile phones in LMICs provides a unique opportunity to harness the potential of mHealth interventions [10]. Mobile WACh (Mobile solutions for Women's and Children's health) is an established human-computer hybrid mobile health (mHealth) system that allows for seamless two-way short message service (SMS) communication with patient tracking capabilities [11, 12]. Mobile WACh or similar mHealth tools are efficient and accessible modalities to connect women and their babies to care [12–16]. Messaging can promote evidence-based interventions including antenatal care attendance, facility delivery and appropriate newborn care while potentially providing real-time, interactive remote support to the women and families [11, 12, 17, 18]. Utilization of this type of SMS messaging has been associated with improved peripartum health knowledge and behaviors, including increased recognition of danger signs [12, 16]. While two-way messaging with a healthcare worker is feasible [12, 19], better understanding end user benefit and the populations best served by these types of intervention is critical to their scalable success and impact on maternal and infant outcomes. We performed a secondary analysis on data from a pilot two-way SMS demonstration project (Mobile WACh NEO pilot) to determine maternal characteristics associated with neonatal death and stillbirth and describe participant messaging to study nurses prior to a stillbirth, infant hospitalization or infant death.

## Methods

This secondary analysis utilized data from the Mobile WACh NEO, a prospective cohort pilot study examining the effect of two-way SMS communication with healthcare workers (HCWs) on neonatal health outcomes. The study was implemented at two public maternal child health (MCH) clinics in Kenya- one in peri-urban Nairobi (Mathare North Health Centre) and one in rural Western Kenya (Rachuonyo Sub-County Hospital).

Pregnant women seeking antenatal care were screened by research staff for participation between May and September, 2017. Women were eligible if they were: ≥14 years of age, between 28–36 weeks estimated gestation, had daily access to a mobile phone, and were able to

communicate via SMS. This gestational age enrollment window was used because most obstetric patients present at these clinics late in the second trimester and the SMS intervention was specifically designed to address the highest risk period just before and after birth in order to potentially have the highest impact and be most feasible [20]. Women who could not read or write independently could participate if they had a trusted person to help them with messages. Participants were counseled by study within a private clinic space and provided written informed consent. Consent counseling was conducted by a study nurse in English, Kiswahili or Dholuo based on participant preference.

Enrolled participants underwent an initial study intake interview assessing sociodemographic data, medical history, and experience with mobile phones. Gestational age was ascertained by last menstrual period (LMP) recorded in the participant's antenatal booklet or fundal height if LMP was not available. All participants were enrolled in an SMS messaging intervention, delivered through a previously described custom web application, Mobile Solutions for Women's and Children's Health (Mobile WACh), designed for semi-automated two-way communication between the participant and an automated system or study staff [11, 12]. This system automatically sent pre-programmed messages in the participant's preferred language and allowed participants to respond at any time. Participant messages were read and responded to by a study nurse during clinic business hours on weekdays. The goal was to respond to every message within the same day if the message was sent during study clinic hours (8AM to 5pm, Monday to Friday). The message system did not capture response times to specific questions participants sent but standard procedures were to reply to urgent messages within an hour or by the end of the day if non-urgent. For those messages received off hours the procedure was to respond in the morning of the next day and Monday morning for weekend messages. The Mobile WACh system also stored and displayed participant characteristics alongside messaging for appropriate responses. SMS were sent from enrollment until 14 weeks postpartum, with a weekly frequency during pregnancy, daily for two weeks postpartum, and twice weekly thereafter. Messages were personalized, action-oriented and included educational messages with advice and a question for participants. Educational messages included topics such as antenatal care, pregnancy complications, birth preparation, newborn care, exclusive breastfeeding, neonatal warning signs, and visit reminders. During enrollment, the study nurse explained that replies to SMS questions were voluntary, however, participants were encouraged to send any questions or concerns. Nurses responded to messages using national guidelines and local practice standards [21, 22]. An OBGYN physician or senior nurse (JU, JK and BW) reviewed messages twice monthly for quality assurance. All messaging was free of charge to participants.

Data collection took place at enrollment (all by in-person visit) and follow-up at 14 weeks postpartum (574 by in-person visit, 83 by phone call). All data collection instruments, including Edinburgh Postnatal Depression Scale [23] and the Abuse Assessment Screen [24] were administered using Open Data Kit by the study nurse [25]. Additionally, study SMS conversations were recorded in the SMS system.

Primary outcomes of this secondary analysis were stillbirth (newborn reported dead at birth) and neonatal death (death between live birth and 28 days of age). Secondary outcomes were infant hospitalization (infant admitted to hospital after initially going home after birth during period of study through 14 weeks of age), infant death (through the end of study at 14 weeks) and perinatal death (including stillbirths and those neonatal deaths up through 6 days of age) [3]. We additionally summarized medical circumstances as reported by the participant in the week preceding stillbirth, infant death or hospitalization. These were ascertained based on participant report through SMS, phone call from study nurse and/or during a 14-week postpartum visit. Participant messaging behavior was described by whether or not they ever

sent a message as well as the number of messages per week in the antenatal period (up to the day prior to delivery).

We evaluated correlates of stillbirth, neonatal death, perinatal death and infant hospitalization. Predictors included a priori defined variables based on literature reviews and hypothesized factors associated with pregnancy and infant outcomes, including parity, multiple gestations, history of pregnancy loss, income and maternal age. Log-binomial regression was used to determine correlates of stillbirth, neonatal death, perinatal mortality and infant hospitalization using a Newey-West estimator. Initial comparison of sociodemographic factors, pregnancy characteristics, and mortality outcomes between study sites identified significant variation. All models were therefore adjusted for site to account for differences by facility. Adjusted relative risks (aRRs) with 95% confidence intervals (CI) were presented.

Human subjects approval was received from Kenyatta National Hospital/ University of Nairobi Ethics Review Committee (P101/02/2017) and the University of Washington Institutional Review Board (STUDY00000526). All methods were performed in accordance with the relevant guidelines and regulations. All participants consented to participate in the study.

## Results

Of 3,108 women screened, 1,089 were eligible and 798 pregnant women were enrolled. Due to one maternal death, follow-up was available on 797 participants. Participant flow diagram shown in Fig 1.

### Participant characteristics

Table 1 shows baseline characteristics and outcomes. Participants enrolled had a median age of 24 years [IQR 21, 29], 94% (747) with eight years of education or more, 87% (683) were married and 37% (293) were primiparous. Of those with previous pregnancies, 24% (122) had a history of pregnancy loss or infant death. Median gestational age at enrollment was 32 weeks [IQR 30, 34] and 1.9% (5) participants had a multiple gestation (twin, etc). Eighty two percent (657) of participants owned their own phone and 91.8% (733) used SMS as a primary mode of communication. Several participant characteristics differed significantly between the rural and peri-urban sites: compared with women at the urban site, those at the rural site had lived farther from the clinic, had higher rates of depression and abuse, lower income and higher rates of unintended pregnancies.

### Birth and infant outcomes

Table 2 shows pregnancy outcomes and messaging behavior. Overall, 98.3% (644/655) of participants delivered in a facility. Median gestational age at birth was 39.4 weeks [IQR 37.9, 41.0] and 18.4% (132) of participants delivered preterm (<37 weeks gestation). During the period of study, 13 stillbirths occurred for a rate of 16 per 1,000 pregnancies, and 17 neonatal deaths occurred for a neonatal mortality of 22 per 1,000 live births. Among the 17 neonatal deaths: 8 (47%) occurred before discharge from facility after birth, 6 (33%) neonates died after discharge home from facility or home birth and 3 (17%) had unclear location and timing of death relative to facility discharge based on available maternal reports. A further 2 infants died at over 28 days of age. (Fig 2) Twenty-eight perinatal deaths (stillbirth or neonatal death up to 6 days of age) occurred for a perinatal mortality of 35 per 1000 pregnancies. Hospitalization occurred in 1.7% of infants.

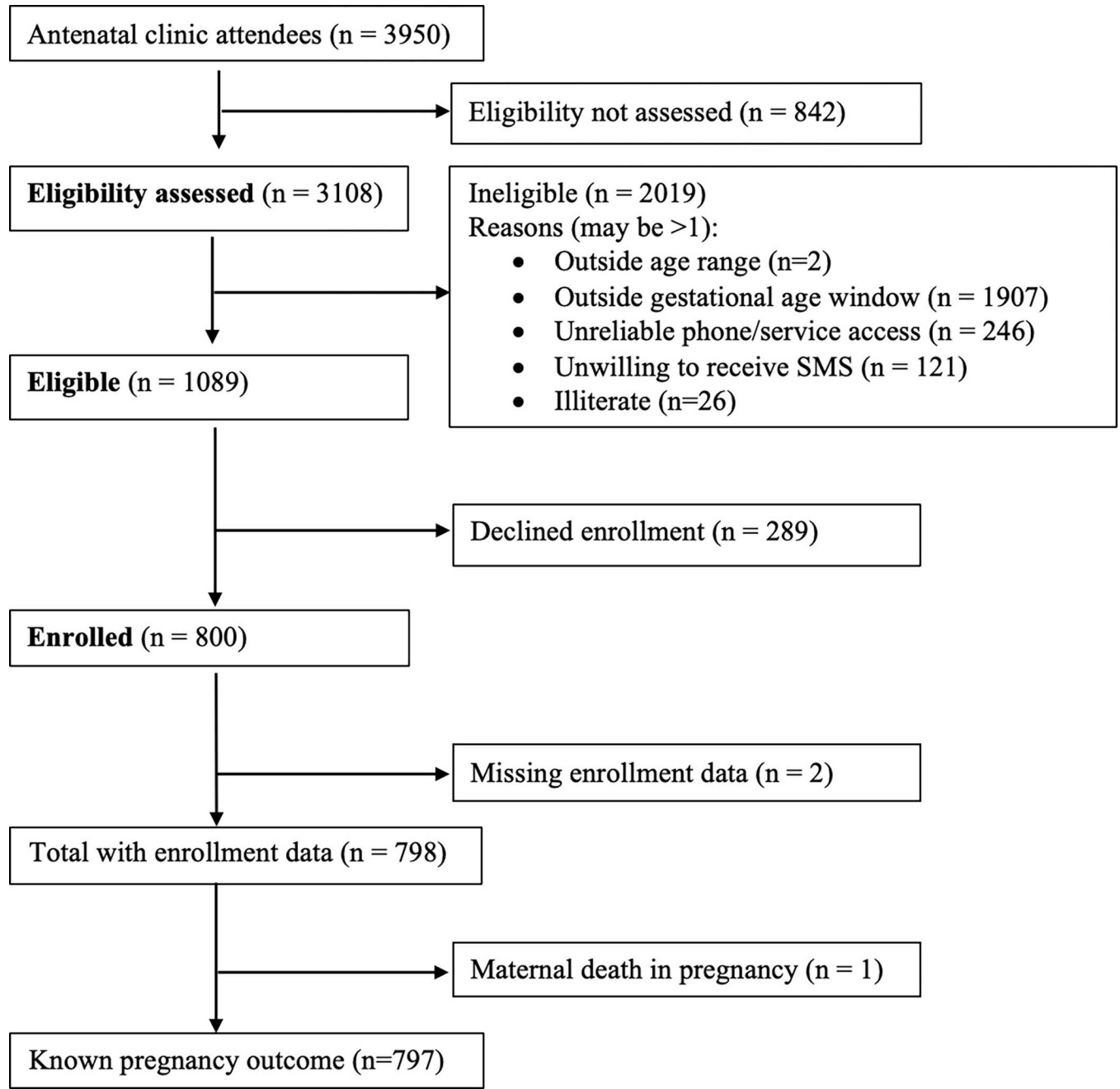

**Fig 1. Diagram of recruitment and enrollment of study participants.** ANC = antenatal care. EDD = estimated due date.

### Risk factors for stillbirth and neonatal death

Associations between participant characteristics and stillbirth and neonatal death are described in Table 3. Maternal age was associated with an increased risk of stillbirth after adjustment for site [aRR 1.12 (CI 1.02, 1.24), $p <0.05$]. Primiparous participants had similar risk of stillbirth to those with a history of prior pregnancy [aRR 0.76 (CI 0.25, 2.38)]. Participants who experienced stillbirth had a trend of sending more messages to the study nurse in the antenatal period than those who did not experience stillbirth; however, this did not reach

**Table 1. Participant characteristics by site.**

| | N | All Participants (N = 797) n (%) or median [IQR] | Rural site (N = 304) n (%) or median [IQR] | Peri-urban site (N = 493) n (%) or median [IQR] |
|---|---|---|---|---|
| Participant Demographics | | | | |
| Maternal age, years | 789 | 24 [21, 29] | 25 [21, 29] | 24 [21, 28] |
| Married | 784 | 683 (87.1%) | 249 (85.3%) | 434 (88.2%) |
| At least eight years of education | 795 | 747 (94.0%) | 278 (91.5%) | 469 (95.1%) |
| Monthly income (thousand KSh) | 761 | 10 [4.5, 14] | 2 [0.5, 7.2] | 11.3 [9, 15]* |
| > 1 hour walk from clinic | 796 | 210 (26.4%) | 133 (43.9%) | 77 (15.6%)* |
| Depression (EDPS score ≥ 13) | 797 | 178 (22.3%) | 122 (40.1%) | 56 (11.4%)* |
| Abuse in last year | 797 | 55 (6.9%) | 29 (9.5%) | 26 (5.3%)* |
| Unintended pregnancy | 797 | 284 (35.6%) | 135 (44.4%) | 149 (30.2%)* |
| Mobile phone and SMS Use | | | | |
| Own own phone | 797 | 656 (82.3%) | 226 (74.3%) | 430 (87.2%)* |
| SMS as primary mode of phone communication | 797 | 733 (91.8%) | 256 (84.2%) | 476 (96.6%)* |
| Pregnancy Characteristics | | | | |
| Primiparity | 797 | 293 (36.8%) | 96 (31.9%) | 197 (40.0%)* |
| Among patients who had a previous pregnancy: | | | | |
| Delivered in health facility | 504 | 410 (81.3%) | 175 (84.1%) | 235 (79.4%) |
| History of pregnancy loss | 504 | 104 (20.6%) | 33 (15.9%) | 71 (24.0%) |
| History of pregnancy or delivery complication | 504 | 48 (9.5%) | 18 (8.7%) | 30 (10.1%) |
| History of infant death | 504 | 50 (9.9%) | 24 (11.5%) | 26 (8.8%) |
| Gestation at enrollment (weeks) | 791 | 32.0 [30, 34] | 32 [28, 34] | 32 [30, 34]* |
| Multiple gestation | 263 | 5 (1.9%) | 2 (1.9%) | 3 (1.9%) |

EDPS = Edinburgh Postnatal Depression Scale

*p < 0.05

statistical significance [median of 1.06 per week vs. 0.78 for a site, aRR 1.4 (CI 0.998, 1.96), *p* = 0.052].

We found no significant predictors of neonatal death; preterm birth was not associated with increased risk of neonatal death [aRR 1.00 (CI 0.98, 1.02)]. We also found no significant predictors of perinatal death and infant hospitalization (data not shown).

## Messaging behavior

Median antenatal exposure time to messaging was 6.7 weeks [IQR 4.4, 9.6]. As described in Table 2, 92% of participants (737) sent at least one message to the study nurse during the study period. Median number of messages sent by participants to study nurse in the antenatal period (up through the day prior to delivery) was 1.0 per week [IQR 0.3, 1.7]. 42.9% (8556/19,950) of all participant messages were sent off hours (outside 8AM to 5PM) or on the weekend. Rural participants messaged less frequently than peri-urban participants: 88% vs. 95% *(p< 0.05)* sent at least one message to the study nurse and a median number of 0.7 vs. 1.1 messages in the antenatal period.

**Stillbirth.** Three participants out of 13 who experienced stillbirths (23%) sent messages to the study nurse describing concerning pregnancy symptoms within 1 week of experiencing stillbirth (Table 4).

**Table 2. Participant messaging behavior and pregnancy outcomes by site.**

| | N | All Participants (N = 797) | Rural site (N = 304) | Peri-urban site (N = 493) |
|---|---|---|---|---|
| | | n (%) or median [IQR] | n (%) or median [IQR] | n (%) or median [IQR] |
| **Messaging Behavior** | | | | |
| Sent at least one message to study nurse | 797 | 737 (92.5%) | 266 (87.8%) | 468 (94.9%)* |
| Antenatal participant messages per week | 797 | 1.0 [0.3, 1.7] | 0.7 [0.1, 1.4] | 1.1 [0.5, 1.8]* |
| Delivery in facility | 655 | 644 (98.3%) | 222 (97.4%) | 422 (98.8%) |
| Neonatal Characteristics (live birth) | | | | |
| Gestation at birth (weeks) | 719 | 39.4 [37.9, 41.0] | 39.4 [37.9, 41.4] | 39.6 [37.7, 40.9]* |
| Preterm birth | 718 | 132 (18.4%) | 49 (18.9%) | 83 (18.1%)* |
| Pregnancy and Infant Outcomes | | | | |
| Neonatal death | 782 | 17 (2.2%) | 5 (1.7%) | 12 (2.5%) |
| Stillbirth | 797 | 13 (1.6%) | 5 (1.6%) | 8 (1.6%) |
| Perinatal death (stillbirths and neonatal death through 6 days of age) | 797 | 28 (3.5%) | 9 (3.0%) | 19 (3.9%) |
| Infant hospitalization | 766 | 13 (1.7%) | 2 (0.7%) | 11 (2.3%) |

*$p < 0.05$

**Hospitalization.** Participants reported 13 infant hospitalizations during the period of study (1.7%); one of these infants died during hospitalization and was accounted for under infant deaths. Four out of the 12 participants (33%) who experienced non-lethal infant hospitalizations messaged the study nurse in regard to the illness before seeking care. (Table 5) Messaging frequency in the antenatal period did not differ between participants who later experienced non-lethal infant hospitalizations (median weekly messages = 1.3 [IQR 0.8, 1.6]) and other participants (median weekly messages = 0.8 [IQR 0.1, 1.6]), aRR 0.96 [CI 0.70, 1.29].

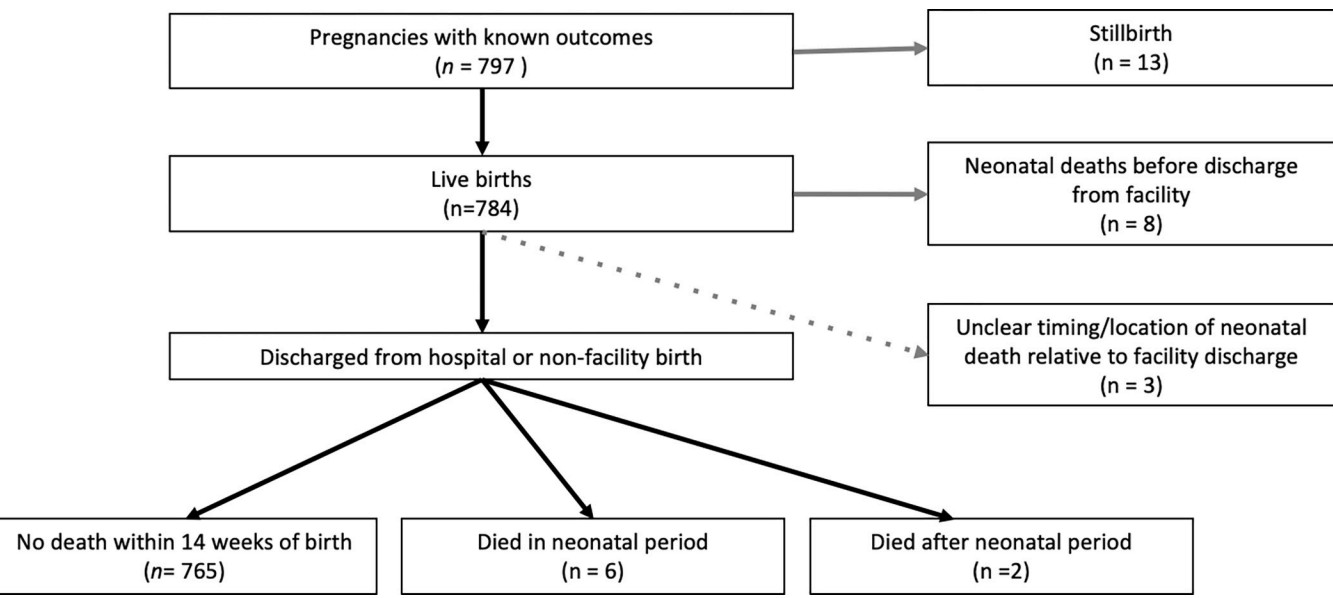

**Fig 2. Flowchart of pregnancy and neonatal outcomes during the period of study.** Neonatal deaths included deaths to 28 days of age (n = 17, neonatal mortality rate of 22 per 1000). Stillbirths (n = 13) occurred for a rate of 16 per 1,000 pregnancies.

**Table 3. Risk factors for neonatal death and stillbirth adjusted for site.**

| | Stillbirth | | | Neonatal death | | |
|---|---|---|---|---|---|---|
| | Stillbirth | Live births | Adjusted RR for site | Neonatal death | No neonatal death^ | Adjusted RR for site |
| | (n = 13) | (n = 784) | | (n = 17) | (n = 765) | |
| | n (%) or median [IQR] | n (%) or median [IQR] | (95% CI) | n (%) or median [IQR] | n (%) or median [IQR] | (95% CI) |
| **Participant Demographics** | | | | | | |
| Maternal age (years) | 28.0 [24.0, 33.0] | 24.0 [21.0, 29.0] | 1.12* (1.02–1.24) | 23.0 [21.0, 27.0] | 24.0 [21.0, 29.0] | 0.95 (0.87–1.04) |
| Married | 12 (92.3%) | 671 (87.0%) | 2.01 (0.25–16.20) | 13 (76.5%) | 657 (87.4%) | 0.51 (0.17–1.57) |
| At least 8 years of education | 12 (92.3%) | 735 (94.0%) | 0.80 (0.11–5.86) | 17 (100%) | 716 (93.8%) | NA |
| Income (1,000 KSh) | 14 [10, 20] | 10 [4, 14] | 1.00 (1.0–1.00) | 11 [8, 13] | 10 [4, 14] | 1.00 (1.00–1.00) |
| > 1 hour walk from clinic | 1 (7.7%) | 209 (26.7%) | 0.21 (0.03–1.54) | 4 (23.5%) | 204 (26.7%) | 0.96 (0.29–3.15) |
| Depression (EDPS ≥ 13) | 5 (38.5%) | 173 (22.1%) | 2.42 (0.86–6.79) | 4 (23.5%) | 168 (22.0%) | 1.30 (0.39–4.38) |
| Abuse in last year | 0 (0%) | 55 (7.0%) | NA | 2 (11.8%) | 52 (6.8%) | 1.94 (0.43–8.77) |
| Pregnancy intended | 12 (92.3%) | 501 (63.9%) | 6.76 (0.82–55.60) | 11 (64.7%) | 489 (63.9%) | 0.97 (0.37–2.54) |
| **Mobile phone and SMS use** | | | | | | |
| Own phone | 13 (100%) | 643 (82.0%) | NA | 13 (76.5%) | 629 (82.2%) | 0.65 (0.21–1.96) |
| SMS primary mode of phone communication | 12 (92.3%) | 720 (91.8%) | 1.07 (0.15–7.89) | 16 (94.1%) | 702 (91.8%) | 1.24 (0.17–8.86) |
| Participant messages per week in antenatal period | 1.06 [0.06, 1.75] | 0.78 [0.09, 1.56] | 1.40 (1.00–1.96) | 0.78 [0.40, 1.17] | 0.78 [0.08, 1.58] | 0.98 (0.65–1.49) |
| **Pregnancy and Neonatal Characteristics** | | | | | | |
| Primiparity | 4 (30.8%) | 289 (36.9%) | 0.76 (0.25–2.38) | 8 (47.1%) | 280 (36.6%) | 1.49 (0.58–3.81) |
| Among participants who had a previous pregnancy: | | | | | | |
| Previously delivered in health facility | 8 (88.9%) | 402 (81.2%) | 0.59 (0.23–1.52) | 7 (77.8%) | 394 (81.2%) | 1.19 (0.77–1.84) |
| History of infant or pregnancy loss | 2 (22.2%) | 120 (24.2%) | 0.96 (0.18–5.23) | 3 (33.3%) | 116 (23.9%) | 1.53 (0.41–5.7) |
| History of pregnancy or delivery complication | 1 (11.1%) | 47 (9.5%) | 1.23 (0.15–10.10) | 2 (22.2%) | 45 (9.3%) | 2.67 (0.56–12.8) |
| Gestation at enrollment (weeks) | 32.0 [30.0, 33.0] | 32.0 [30.0, 34.0] | 1.00 (1.00–1.00) | 32.0 [30.0, 34.0] | 32.0 [29.0, 34.0] | 1.00 (1.00–1.01) |
| Multiple gestation | 1 (16.7%) | 4 (1.6%) | 8.73 (0.89–85.40) | 0 (0%) | 162 (25.8%) | NA |
| Gestation at birth (weeks) | 38.2 [36.6, 40.2] | 39.0 [37.0, 40.7] | 1.00 (1.00–1.00) | 38.7 [37.2, 39.7] | 39.0 [37.0, 40.7] | 1.00 (1.00–1.00) |
| Preterm birth (delivery before 37 weeks) | 3 (30.0%) | 165 (25.7%) | 1.00 (0.98–1.03) | 3 (23.1%) | 162 (25.8%) | 1.00 (0.98–1.02) |

*p<0.05.

^2 infants died after neonatal period (>28 days) but before study ended and were not included. EDPS = Edinburgh Postnatal Depression Scale

**Table 4. Description of SMS interactions describing concerning pregnancy symptoms within 1 week of stillbirth.**

| # | Excerpt from participant message | Nurse intervention/ Stillbirth description |
|---|---|---|
| 1 | "...for the last 2 days am not hearing my baby playing is something wrong or is normal". | Participant was advised by nurse to go to a facility where no fetal heart rate was found on scan. |
| 2 | "the problem is I have high blood pressure which is worrying me soo much" | Nurse confirmed participant had gone to the hospital for a checkup, had medication and knew about dietary modifications that can be helpful. Participant was later told by medical provider she delivered stillborn due to high blood pressure. |
| 3 | "...The only problem I have is the weight am carrying, too heavy" | One of the twins was stillborn, participant was told by medical provider her problem was "swollen legs". |

**Table 5. Description of SMS interactions describing infant symptoms in week prior to hospitalization.**

| # | Excerpt from participant message | Nurse intervention/ Hospitalization description |
|---|---|---|
| 1 | "...the baby is not breathing the chest is blocked" | Nurse confirmed patient was taken to hospital. Infant was hospitalized for difficulty breathing |
| 2 | "...the cord has been dry all along but in the afternoon i've seen some wetness and the baby is crying what should i do?" | The nurse asked follow up questions but did not hear back, patient admitted three days later for neonatal sepsis and dehydration. |
| 3 | "...the baby is not feeling well" | Nurse pressed participant to take baby to the hospital. Patient admitted the following day and treated for jaundice and clavicle fracture. |
| 4 | "the baby has a fhigh body temeperature na flu too" | Baby taken to hospital and treated for pneumonia. |

**Infant death.** Nineteen infant deaths (including 17 during the neonatal period) were reported. Eight of these deaths occurred in the hospital after delivery. Our analysis focused on the remaining eleven of these deaths that occurred after the infant had gone home from facility birth or had unclear timing of death, as these may be preventable with improved connection to medical care. The available information describing these infants is in Table 6. Two newborns were preterm, five deaths occurred in or on the way to a facility and presenting signs of terminal illness included: respiratory distress (4/11), fever (2/11), poor feeding (2/11) and unknown (3/11). Two participants sent messages to study nurses within one week of infant death. Only

**Table 6. Description of circumstances and SMS interactions for mothers whose infant died after time at home.**

| # | Wks gestation | Age in days at death | Location/ timing of death | Presenting sign of terminal illness | SMS from mother near infant death | Available description of circumstances surrounding death |
|---|---|---|---|---|---|---|
| 1 | 40 | 0 | At home following delivery | Respiratory distress | None | Vaginal delivery at home, neonate died due to birth asphyxia. |
| 2 | 34* | 0 | Unclear | Unknown | None | Birth at 34 weeks, no other information |
| 3 | 40 | 1 | Unclear | Unknown | None | No information provided by mother. |
| 4 | 38 | 2 | In transport to facility | Fever | Day prior to death: "milk was not coming out yesterday but now the baby is breast feeding well..." | Neonate had inconsolable crying, failure to feed and high fever- died on the way to the hospital. |
| 5 | 39 | 2 | Unclear | Unknown | None | Vaginal delivery and neonate died at 2 days of age. |
| 6 | 40 | 4 | In transport to facility | Fever | None | Neonate developed fever and died on the way to hospital. |
| 7 | 37 | 6 | At home | Respiratory distress | None | Mother noticed difficulty breathing and gasping sounds at 3am, neonate stopped breathing a few minutes later. |
| 8 | 39 | 8 | In transport to facility | Respiratory distress | None | Neonate developed difficulty breathing at 7 days of age and died on arrival to the hospital the following day. |
| 9 | 35* | 14 | At home | Poor feeding | None | Neonate refused to breastfeed for two days following birth and died at 14 days with severe dehydration. |
| 10 | 42 | 49 | During subsequent hospitalization | Poor feeding | "My baby has something like a boil on the cheek this evening" | Infant with boil and refusal to breastfeed, taken to hospital and died two days later with diagnosis of severe dehydration. |
| 11 | 41 | 57 | In transport to facility | Respiratory distress | None | Infant developed difficult breathing and started "losing breath"- died while mother was rushing her to the hospital. |

All data from maternal report via SMS or in phone/in person interview. *Preterm (gestation <37 weeks)

one of these messages described a concerning health symptom, a boil on the cheek. A total of 23% (3/13) and 33% (4/12), respectively, of participants who subsequently had a stillbirth or infant hospitalization messaged the study nurse with a concern within a week of the event. In contrast, however, only 9% (1/11) of participants whose infant died messaged the study nurse within the week preceding the event.

Eighty two percent (657/797) of study participants had an end of study visit and answered questions in regard to their messaging behavior. This included 8 out of 11 participants who experienced an infant death after discharge home. Half (4/8) of these reported they consulted the nurse via SMS during the study vs. 72% (454/630) of participants without infant death (*p = 0.3*).

## Discussion

This secondary analysis identified increasing maternal age as a risk factor for antepartum stillbirth, as well as described reports from mothers about stillbirths and infant deaths in a demonstration project of SMS messaging in Kenya. This study additionally provided insights into SMS messaging behavior around the time of these adverse events.

While we identified maternal age as a risk factor for stillbirth, no other factors were associated with neonatal death or perinatal death. Advanced maternal age has previously been associated with higher risk of perinatal mortality in Kenya and elsewhere [5, 26]. Other established risk factors for stillbirth and neonatal death, such as preterm birth, primiparous status and history of pregnancy loss [27–29] trended but did not reach significance, which may have been affected by sample size. Importantly insufficient data was available for perinatal conditions associated with stillbirth such as diabetes and preeclampsia. We did not abstract data from clinical records and documentation on maternity booklets was limited.

Primary outcomes in our study of stillbirth (16 per 1,000 pregnancies) and perinatal death (36 per 1,000 pregnancies) were above Kenyan national rates of 13 and 29 respectively [20]. The rates may be higher than national rates due to closer monitoring for pregnancy outcomes in a research study. The study neonatal mortality (22 per 1,000 live births) was equal to Kenyan national rates [20].

Among the eleven mothers who experienced an infant death after discharge home, only one messaged the study nurse with concerns about their baby in the week preceding death. To our knowledge prior studies have not reported about mHealth messaging to study personnel around these adverse events. While many SMS programs for maternal and child health exist in Kenya and other LMIC, few offer two way messaging with a healthcare provider and fewer have been evaluated for impact on health outcomes. In addition, most mHealth interventions such as this are not designed to respond to participant's urgent health needs. The mothers in this project appeared to be equally "connected" as those who did not experience infant death as shown by their equivalent baseline messaging frequency in the antenatal period. Despite previous messaging by these mothers, they did not seek support from the SMS system during their infants' acute illness. This may be because many of the deaths were likely due to neonatal sepsis, which is a rapidly progressing entity and limits mothers' ability to message the study nurse. In addition, the number of infants who died after discharge from the hospital is small and therefore limits the conclusions that can be made about the SMS messaging. However, it does suggest that more investigation needs to be done to understand how mHealth interventions may support families in times of more critical illness. Of note, almost a quarter of women who experienced a stillbirth did text during the week prior to the identification of the stillbirth. Stillbirth, while preventable in some cases, remains very difficult to predict, and often women do not experience identifiable signs and symptoms that could be addressed by the Mobile WACh counseling.

In contrast to infrequent messaging among mothers preceding infant death, one third of participants messaged the study nurse with concerns within a week of their infant's non-lethal hospitalization. These mothers may have had more time to message given less severe illness presentations in their babies, and/or utilization of the message system may have helped them identify an illness early that was able to be appropriately treated in a facility and avoid death. However, a larger trial is needed to understand this impact.

The Mobile WACh Neo demonstration project was a novel use of a highly available, low cost modality to communicate with, educate and support pregnant women and mothers of newborns on a weekly and then daily basis. Strengths of this secondary analysis includes use of detailed data on sociodemographic and potential risk factors as well as outcome data on the majority of participants. Limitations of our study include the lack of a control group and therefore unknown comparative outcomes if participants did not receive messaging. Additionally, all outcomes were subject to maternal report, which limited information available about timing, neonatal cause of death or hospitalization.

## Conclusion

In this study we found only increasing maternal age was a predictor for stillbirth among women living in areas with high rates of stillbirth, perinatal and infant mortality. Other demographic, birth outcomes and obstetrical factors were not associated with these adverse events. This suggests that women even without previously associated risk factors experience stillbirth and infant loss and therefore very specific targeted interventions will miss these women and their babies. The goal of Mobile WACh NEO pilot was to assist women in identifying danger signs around pregnancy and infant health. While women were very engaged in the program and study nurses received timely communication by some participants with impending stillbirth or neonatal hospitalization most of the few women who experienced acute infant illness after going home did not engage with the nurse during this critical time prior to the event. Low levels of messaging in the acute period from mothers whose infant subsequently died after discharge home may reflect the early timing of neonatal deaths and difficulty in utilizing a study nurse during this intense emergency period. These babies went home and appeared to get sick fast, generally in the first week of life and with symptoms that presented quickly including overnight fevers and breathing difficulty. Therefore, effective interventions via two way SMS for this high-risk neonatal period may need to focus primarily on antenatal anticipatory guidance to engage women and families in early identification of neonatal illness when it later occurs and focus on connecting women to care quickly and at all times. Our ongoing randomized controlled study (Mobile WACh NEO RCT) incorporates these approaches and is in progress to determine the efficacy of this strategy to protect newborns and inform possible scale up of these programs.

## Author Contributions

**Conceptualization:** Brenda Wandika, Maneesh Batra, Dalton Wamalwa, Grace John-Stewart, John Kinuthia, Jennifer A. Unger.

**Data curation:** Lusi Osborn.

**Formal analysis:** Anna B. Hedstrom, Esther M. Choo, Keshet Ronen, Wenwen Jiang, Lusi Osborn, Jennifer A. Unger.

**Funding acquisition:** Grace John-Stewart, John Kinuthia, Jennifer A. Unger.

**Methodology:** Keshet Ronen, Brenda Wandika, Lusi Osborn, Maneesh Batra, Dalton Wamalwa, Grace John-Stewart, John Kinuthia, Jennifer A. Unger.

**Project administration:** Brenda Wandika, John Kinuthia, Jennifer A. Unger.

**Supervision:** Brenda Wandika, Maneesh Batra, Grace John-Stewart, John Kinuthia, Jennifer A. Unger.

**Validation:** Brenda Wandika, Jennifer A. Unger.

**Visualization:** Anna B. Hedstrom, Brenda Wandika.

**Writing – original draft:** Anna B. Hedstrom.

**Writing – review & editing:** Esther M. Choo, Keshet Ronen, Brenda Wandika, Maneesh Batra, Dalton Wamalwa, Grace John-Stewart, John Kinuthia, Jennifer A. Unger.

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
