## [Decision Letter · Decision Letter 0]

18 Feb 2022

PGPH-D-21-00877

Risk Factors for Stillbirth and Neonatal Mortality Among Participants in Mobile WACh NEO, a Pilot Two-Way SMS Communication Program in Kenya

Dear Dr. Hedstrom,

Thank you for submitting your manuscript to PLOS Global Public Health. After careful consideration, we feel that it has merit but does not fully meet PLOS Global Public Health’s publication criteria as it currently stands. Therefore, we invite you to submit a revised version of the manuscript that addresses the points raised during the review process.

We look forward to receiving your revised manuscript.

Kind regards,

Bethany Hedt-Gauthier, PhD

Academic Editor

Journal Requirements:

1. Please amend your detailed Financial Disclosure statement. This is published with the article, therefore should be completed in full sentences and contain the exact wording you wish to be published.

i). Please include all sources of funding (financial or material support) for your study. List the grants (with grant number) or organizations (with url) that supported your study, including funding received from your institution. 

ii). State the initials, alongside each funding source, of each author to receive each grant.

iii). If any authors received a salary from any of your funders, please state which authors and which funders.

2. Please ensure that the funders and grant numbers match between the Financial Disclosure field and the Funding Information tab in your submission form. Note that the funders must be provided in the same order in both places as well.

3. Please remove your Figure 1 legend from the Supporting Information section of your manuscript.

Additional Editor Comments (if provided):

The reviewers have provided thoughtful and extensive comments for you to address. As you consider the comment about the inclusion of co-authors from Kenya, I encourage you to review the newest authorship reflexivity statements (https://associationofanaesthetists-publications.onlinelibrary.wiley.com/doi/10.1111/anae.15597) and use this as a guide for that response.

Reviewers' comments:

Reviewer's Responses to Questions

**Comments to the Author**

1. Does this manuscript meet PLOS Global Public Health’s publication criteria? Is the manuscript technically sound, and do the data support the conclusions? The manuscript must describe methodologically and ethically rigorous research with conclusions that are appropriately drawn based on the data presented.

Reviewer #1: Partly

Reviewer #2: Yes

Reviewer #3: Partly

2. Has the statistical analysis been performed appropriately and rigorously?

Reviewer #1: Yes

Reviewer #2: Yes

Reviewer #3: Yes

3. Have the authors made all data underlying the findings in their manuscript fully available (please refer to the Data Availability Statement at the start of the manuscript PDF file)?

Reviewer #1: Yes

Reviewer #2: No

Reviewer #3: Yes

4. Is the manuscript presented in an intelligible fashion and written in standard English?

Reviewer #1: No

Reviewer #2: Yes

Reviewer #3: Yes

5. Review Comments to the Author

Reviewer #1: This manuscript would be a helpful contribution to the literature for understanding how mhealth could help to increase access to care for pregnant women and mothers in post-partum for improved and maternal outcomes. I hope the authors will find the comments below helpful as they revise the paper.

Background:

1) Lines 75-78, authors should be clear about this study aim/research question. Do you aim at assessing the effect of the “two-way SMS intervention” during pregnancy and post-partum on maternal and infant outcomes? Or your objective was just to describe these outcomes among the study participants?

Methods’ section:

2) As you describe the intervention, include specific information about the expected speed of responding to the messages from mothers by study nurses. You should also summarize this information in the results’ section, as well as the number of messages sent during the non-working hours or the weekend.

3) You should include a flowchart of the recruitment of study participants, with a brief description of exclusion/inclusion criteria on each step.

a. Lines (134-135), you said that of the 3,108 women screened, 1,089 were eligible for this study, but only 798 were enrolled – what happened to the other 291 eligible women?

4) For the study inclusion criteria, you should explain the rationale of choosing pregnant women at 28-36 weeks of gestation. There is evidence that early initiation of antenatal care (<12 weeks gestation) is associated with improved pregnancy and birth outcomes, why not taking this window of opportunity?

5) You will need to revise the statistical analysis part according to the study objectives. If the study aim was to assess the effect of exposure to the “two-way SMS intervention” on maternal and infant outcomes, then the expected analysis should be assessing whether there was an association between exposure to the intervention and outcomes, controlling for possible confounding factors, including socio-demographic characteristics.

Results’ section:

6) In Tables 1&2, you should include at the top the total number of study participants (N) for each study site (rural/urban).

Discussion section:

7) The discussion section should be deeply revised to reflect more of what should be expected from a classical discussion section:

a) Include an introductory as the first paragraph for summarizing the study aim and all key findings.

b) Interpret the key findings by comparing them with the previous literature

c) Discuss the major limitations of this study, and how they should affect the interpretation of the findings. For example,

d) Discuss the implications of your results as well/conclusions from your findings.

8) There is no need to put much effort in comparing your study estimates to the other estimates reported by nationally-representative surveys. Your study sample should never be expected to produce nationally-representative estimates. However, you may want to discuss any comparisons under the study limitations.

9) Lines 232-240, again, why didn’t you adopt the WHO’s definition of stillbirth when you defined the study inclusion criteria?

10) Lines 263-265, I don’t understand the following statement and it is not suggested by your findings. “Our results suggest that prenatal identification of populations at high risk of perinatal death may not be an optimal method for targeting mHealth programs, and much like antenatal care programs they should instead be available to all pregnant women”.

a. Make sure that your conclusions are supported by the study results.

Reviewer #2: Thank you for inviting me to review this interesting research work in two public clinic in Nairobi, Kenya. This study aims to see the effect of introduction of mobile based communication to women during pregnancy and is definitely valuable in providing new evidence on effect of innovation on health care.

However, there are some major concerns in terms of intervention introduction process.

First, any innovation cannot be introduced in vacuum, it requires context, facilitator for successful implementation. Please see this important reference https://implementationscience.biomedcentral.com/articles/10.1186/1748-5908-7-25. The researcher have not provided how the implementation was done. Who were the facilitators to introduce the innovation, what was the process like.

Second, the flow figure doesnot cover the STROBE flow diagram. The 3,108 women screened and 1,089 eligible were not mentioned in the flow figure.

Third, this study aims to provide the association of use of two way communication with the health outcomes. The researcher need to characterise the population who used the two way communication and those who didnot use two way communication using the Mobile WACh. The table 1 and 2 is redundant.

Fourth, the causal pathway i.e the DAG mapping needs to be explained in the method. The expectation here is the two way communication effects on health outcome mental and mortality, I think there are a number of social, biological and health service factors which effect the mental health and mortality. This is not provided a causal pathway and should be presented as multi-level analysis.

Fifth, I see most of the authors from US and only two from Kenya. Being a researcher and an advocate for decolonize global health, I think there should be a right balance of authorship. We need to promote researchers from global South.

Sixth, this study has merit, but needs a revision of tables as suggested above.

Reviewer #3: Thank you for submitting this well-written article. The study provides some interesting findings as it relates to the use and implications of mobile technology in improving birth outcomes, but the manuscript could benefit from an expansion of the literature review, as well as more details about the implications of the study as it relates to policy.

6. PLOS authors have the option to publish the peer review history of their article (what does this mean?). If published, this will include your full peer review and any attached files.

**Do you want your identity to be public for this peer review?** For information about this choice, including consent withdrawal, please see our Privacy Policy.

Reviewer #1: **Yes: **Alphonse Nshimyiryo

Reviewer #2: No

Reviewer #3: No

---

## [Decision Letter · Decision Letter 1]

17 May 2022

PGPH-D-21-00877R1

Risk Factors for Stillbirth and Neonatal Mortality Among Participants in Mobile WACh NEO Pilot, a Two-Way SMS Communication Program in Kenya

Dear Dr. Hedstrom,

Thank you for submitting your manuscript to PLOS Global Public Health. After careful consideration, we feel that it has merit but does not fully meet PLOS Global Public Health’s publication criteria as it currently stands. Therefore, we invite you to submit a revised version of the manuscript that addresses the points raised during the review process.

We look forward to receiving your revised manuscript.

Kind regards,

Bethany Hedt-Gauthier, PhD

Academic Editor

Journal Requirements:

Additional Editor Comments (if provided):

Reviewers' comments:

Reviewer's Responses to Questions

**Comments to the Author**

1. If the authors have adequately addressed your comments raised in a previous round of review and you feel that this manuscript is now acceptable for publication, you may indicate that here to bypass the “Comments to the Author” section, enter your conflict of interest statement in the “Confidential to Editor” section, and submit your "Accept" recommendation.

Reviewer #1: All comments have been addressed

Reviewer #2: All comments have been addressed

Reviewer #3: (No Response)

2. Does this manuscript meet PLOS Global Public Health’s publication criteria? Is the manuscript technically sound, and do the data support the conclusions? The manuscript must describe methodologically and ethically rigorous research with conclusions that are appropriately drawn based on the data presented.

Reviewer #1: Yes

Reviewer #2: Yes

Reviewer #3: Yes

3. Has the statistical analysis been performed appropriately and rigorously?

Reviewer #1: Yes

Reviewer #2: Yes

Reviewer #3: Yes

4. Have the authors made all data underlying the findings in their manuscript fully available (please refer to the Data Availability Statement at the start of the manuscript PDF file)?

Reviewer #1: Yes

Reviewer #2: Yes

Reviewer #3: Yes

5. Is the manuscript presented in an intelligible fashion and written in standard English?

Reviewer #1: Yes

Reviewer #2: Yes

Reviewer #3: No

6. Review Comments to the Author

Reviewer #1: Again, thank you for the opportunity to review the revised version of this paper. I would like to reiterate that this study will be a helpful contribution to the literature for understanding how mhealth could help to increase access to care for pregnant women and mothers in post-partum for improved and maternal outcomes. Authors have addressed most of my comments on the previous version, however below are a couple of minor comments on the current version:

1) For the study inclusion criteria, it would be important to explain the rationale of enrolling in this study pregnant women at 28-36 weeks of gestation. There is evidence that early initiation of antenatal care (<12 weeks gestation) is associated with improved pregnancy and birth outcomes, why not taking this window of opportunity? This comment on the previous version remains not addressed.

2) Lines 247-248. I think the following statement “Given the median enrollment age of 32 weeks, not all participants were monitored starting at the WHO defined stillbirth range of 28 weeks” doesn’t help to explain the discrepancy between the stillbirth/neonatal rates in this study and the national rates. The enrollment age shouldn’t affect the still birth rates, as the study only included pregnant women and followed up all of them to assess their outcomes and the study doesn’t report a significant loss-to-follow-up.

Reviewer #2: Dear Authors

There have been much improvement in the revision. I still have some comments that the authors need to consider before final decision

1. This is a cohort study, so please use the standard epidemiological language rather than longitudinal study.

2. One of the key finding is that the proportion women who had stillbirth messaged less frequently than women who had live birth. This relates well to the "Thaddeus and Mine" three delay model https://doi.org/10.1016/S0277-9536(97)10018-1. Women did not recognize the danger signs as a result there was delay in care. The NeoWatch technology helped to identify this important barrier to care. In that aspect, improving the use of the NeoWatch needs further contextualising the intervention. This needs to be reflected both in abstract and discussion.

3. This still is a epidemiological study as you compared the risk factor for stillbirth and used messaging as exposure, so please use the STROBE checklist. CONSORT checklist is used in intervention study particularly if randomization to intervention is done.

Reviewer #3: I have responded to the PLOS editorial questions above. have no additional comments

7. PLOS authors have the option to publish the peer review history of their article (what does this mean?). If published, this will include your full peer review and any attached files.

**Do you want your identity to be public for this peer review?** For information about this choice, including consent withdrawal, please see our Privacy Policy.

Reviewer #1: **Yes: **Alphonse Nshimyiryo

Reviewer #2: No

Reviewer #3: No

---

## [Editor Report · Decision Letter 2]

29 Jun 2022

Risk Factors for Stillbirth and Neonatal Mortality Among Participants in Mobile WACh NEO Pilot, a Two-Way SMS Communication Program in Kenya

PGPH-D-21-00877R2

Dear Dr. Hedstrom,

We are pleased to inform you that your manuscript 'Risk Factors for Stillbirth and Neonatal Mortality Among Participants in Mobile WACh NEO Pilot, a Two-Way SMS Communication Program in Kenya' has been provisionally accepted for publication in PLOS Global Public Health.

Best regards,

Bethany Hedt-Gauthier, PhD

Academic Editor
